# Clinical Features and Management of Acute and Chronic Radiation-Induced Colitis and Proctopathy

**DOI:** 10.3390/cancers15123160

**Published:** 2023-06-12

**Authors:** Hamzah Abu-Sbeih, Tenglong Tang, Faisal S. Ali, Weijie Ma, Malek Shatila, Wenyi Luo, Dongfeng Tan, Chad Tang, David M. Richards, Phillip S. Ge, Anusha S. Thomas, Yinghong Wang

**Affiliations:** 1Department of Gastroenterology, Hepatology, and Nutrition, The University of Texas MD Anderson Cancer Center, Houston, TX 77030, USA; hamzah_abusbeih@hotmail.com (H.A.-S.); tangtenglong@csu.edu.cn (T.T.); jeffreyeah@163.com (W.M.); mshatila@mdanderson.org (M.S.); dmrichards@mdanderson.org (D.M.R.); psge@mdanderson.org (P.S.G.); asthomas1@mdanderson.org (A.S.T.); 2Department of Internal Medicine, University of Missouri, Kansas City, MO 65211, USA; 3Department of General Surgery, The Second Xiangya Hospital of Central South University, Changsha 410011, China; 4Department of Gastroenterology, Hepatology, and Nutrition, The University of Texas Health Science Center, Houston, TX 77030, USA; faisalshaukatali@gmail.com; 5Department of Hepatobiliary and Pancreatic Surgery, Zhongnan Hospital of Wuhan University, Wuhan 430071, China; 6Department of Pathology, The University of Texas MD Anderson Cancer Center, Houston, TX 77030, USA; wenyi-luo@ouhsc.edu (W.L.); dtan@mdanderson.org (D.T.); 7Department of Radiation Oncology, The University of Texas MD Anderson Cancer Center, Houston, TX 77030, USA; ctang1@mdanderson.org

**Keywords:** radiation, colitis, proctopathy, endoscopy, argon plasma coagulation

## Abstract

**Simple Summary:**

Radiation-induced colitis and proctopathy (RICAP) is a recognized adverse effect of radiation therapy (RT) that can adversely affect cancer patients’ quality of life. However, data on the clinical characteristics and outcomes of RICAP are scarce. Our study found that acute RICAP (ARICAP) has non-bloody diarrhea as the predominant symptom, whereas chronic RICAP (CRICAP) has mostly bloody diarrhea. ARICAP patients more often received medical management, whereas CRICAP patients with bleeding more often received argon plasma coagulation (APC). APC treatment may be useful in patients with endoscopically apparent bleeding, but it did not significantly reduce the bleeding recurrence or RICAP symptoms. More research is needed to better characterize and distinguish between the two entities.

**Abstract:**

Background: RICAP is a recognized adverse effect of radiation therapy (RT) that can adversely affect cancer patients’ quality of life. Data on the clinical characteristics and outcomes of RICAP are scarce. We aimed to analyze the clinical and endoscopic characteristics of acute or chronic radiation-induced colitis and proctopathy (ARICAP and CRICAP) based on symptom onset after RT (≤ or >45 days, respectively). Methods: This is a retrospective observational study of a single tertiary cancer center, from January 2010 and December 2018, of cancer patients with endoscopically confirmed ARICAP and CRICAP. We conducted univariate and multivariate logistic regression analyses to associate clinical variables with endoscopic and medical outcomes. Results: One hundred and twelve patients were included (84% Caucasian; 55% female; median age of 59 years); 46% had ARICAP with non-bloody diarrhea as the predominant symptom, whereas 55% had CRICAP with mostly bloody diarrhea. Neovascularization was the most frequent finding on endoscopy, followed by bleeding. ARICAP patients more often received medical management (*p* < 0.001), whereas CRICAP patients with bleeding more often received argon plasma coagulation (APC) (*p* = 0.002). Female sex and undergoing less-intense RT treatments were more associated with medical treatment; bleeding clinically and during the endoscopy was more associated with APC treatment. However, APC treatment did not significantly reduce bleeding recurrence or RICAP symptoms. Conclusion: Patients with ARICAP and CRICAP experience different symptoms. Medical management should be considered before endoscopic therapy. APC may be useful in patients with endoscopically apparent bleeding.

## 1. Introduction

Radiation-induced colitis and proctopathy (RICAP) is known to develop in patients undergoing lower abdominal radiotherapy (RT) for genitourinary, gynecological, or gastrointestinal (GI) malignancies [1]. Two distinct types of RICAP are clinically recognized: acute RICAP (ARICAP), which develops within 6 weeks of RT, and chronic RICAP (CRICAP), which has a more delayed onset after RT and may take up to a year to manifest clinically [2].

The incidence of RICAP is estimated at 2–20%, with some studies suggesting incidences as high as 40%, depending on multiple factors such as pre-existing inflammatory bowel disease, the type of radiotherapy administered (brachytherapy, external beam radiation, etc.), and radiotherapy-specific parameters [3,4]. Specifically, the duration, distribution, and volume of radiation exposure as well as the fractionation, dose, and intensity of ionizing radiation all play an important role in the ensuing inflammatory process [5].

The treatment of RICAP relies on medical management with antidiarrheal agents, corticosteroid suppositories or enemas, and fluid resuscitation [6]. Interventional endoscopic techniques, which have advanced substantially in recent years, can be used in select cases. Argon plasma coagulation (APC), which has replaced traditionally used bipolar devices, is the most-used endoscopic intervention for RICAP-associated bleeding, given its limited depth of coagulation (0.5–3 mm) [7,8]. However, given limited data on the clinical features, endoscopic characteristics, management, and outcomes of RICAP, our understanding of ARICAP and CRICAP remains inadequate. The aim of our study was to assess the clinical characteristics and outcomes of ARICAP and CRICAP and assess the utility of endoscopic intervention in their management.

## 2. Materials and Methods

### 2.1. Study Design and Population Characteristics

This was a retrospective, descriptive, single-center study of cancer patients who received RT to the lower abdomen or pelvis and underwent endoscopic evaluation between January 2010 and December 2018. For the purpose of this study, patients were deemed to have RICAP if they developed either rectal pain, hematochezia, or tenesmus in addition to other lower GI symptomatology (diarrhea, abdominal pain, etc.) [9]. We included patients who (1) were 18 years or older at the time of receiving RT for a confirmed malignancy; (2) had RT to the lower abdomen or pelvis; (3) developed symptoms of RICAP; and (4) underwent lower endoscopy to confirm RICAP. Patients with infectious/inflammatory colitis, chemotherapy/immunotherapy-induced colitis, and graft versus host disease or no endoscopic evaluation were excluded from the study.

Data extracted from patients’ medical records and the pharmacy database included patients’ demographics, concomitant morbidities, cancer-related information, and baseline medication use within 3 months of RT.

### 2.2. Oncological Data and Treatment Courses

Malignancies were categorized as prostate, cervical, anal, recto-sigmoid (including all rectal and sigmoid malignancies), or miscellaneous (i.e., ovarian, pelvic bone, and vaginal). The cancer stage was assessed according to the American Joint Committee on Cancer Staging System, 7th edition. Use of chemotherapy known to cause GI toxicity within 6 months of RT was also recorded. RT was classified as external beam (intensity-modulated RT; proton, photon, electron, or 3-dimensional conformal RT) or brachytherapy (as monotherapy or in combination with other therapies). As this study focuses primarily on clinical features, only data regarding radiotherapy duration and dosage were included.

### 2.3. RICAP Data

RICAP was categorized as ARICAP (occurring during RT or ≤45 days after last RT) and CRICAP (occurring > 45 days after last RT) [10]. Symptoms were graded based on the common terminology criteria for adverse events (CTCAE), version 5.0 [11]. Data on RICAP medical treatments, hospitalizations, and recurrences were extracted.

### 2.4. Clinical Success and Endoscopic and Pathologic Features

We extracted the following endoscopic findings: presence of ulceration, stricture, neovascularization, and bleeding. RICAP was classified as rectal or non-rectal (the rest of the colon). The nature (initial or repeated) and number of APC treatments were recorded. The pathological characteristics of RICAP were also reviewed and reported. We defined successful management of RICAP as lack of recurrent symptoms after treatment of the index event.

### 2.5. Statistical Analysis

We used SPSS version 24.0 (IBM Corporation, Armonk, NY, USA) for statistical analysis. The distribution of categorical variables was summarized by percentages and frequencies, and that of continuous variables by medians and interquartile ranges (IQRs) or means and standard deviations. Categorical variables were compared using the Fisher exact test. Continuous variables were compared using the Wilcoxon rank sum test. Univariate logistic regression analysis was conducted to assess the risks of complications, treatment requirement, and visible bleeding on endoscopy. All statistical evaluations were 2-sided, and *p* values less than 0.05 were considered statistically significant.

## 3. Results

### 3.1. Patients’ Baseline Demographics and Cancer-Related Variables

A total of 61,399 patients who underwent RT of the abdomen and pelvis within the study period were identified. Of these, 7185 underwent endoscopy for suspected RICAP, and 112 patients had confirmed RICAP (84% Caucasian; median age, 59 years (IQR, 50–68); 55% female) (Table 1).

Overall, 46% of patients had ARICAP, and 55% had CRICAP. The most common underlying malignancy was prostate cancer (33%). Only 5% of the patients had stage IV malignancies. Half (50%) of the patients underwent chemotherapy with a notable GI toxicity profile within 6 months of RT. The median RT dose was 55 Gy (IQR, 50–70), and RT was administered over a median of 30 sessions (IQR, 26–35) and a median of 43 days (IQR, 37–51). Overall, 18% of patients received brachytherapy. Over a median follow-up of 10 years (IQR, 5–13), 38% of patients had non-GI adverse events associated with RT.

### 3.2. RICAP Clinical Features

The median time from the first RT session to the colitis onset was 14 days (IQR, 7–21) for patients with ARICAP and 404 days (IQR, 183–6895) for those with CRICAP (Table 2). CRICAP was significantly associated with a higher number of radiotherapy sessions (median 35 sessions vs. 28 in ARICAP; *p* < 0.001) and a higher dose of radiation (median 66 Gy vs. 54 in ARICAP; *p* < 0.001). Non-bloody diarrhea was reported in 92% of patients with ARICAP, whereas 92% of patients with CRICAP had bloody stools. Additionally, 45% of patients with ARICAP, but only 2% of patients with CRICAP experienced nausea and vomiting. The median duration of symptoms was shorter in ARICAP than in CRICAP patients (16 vs. 29 days, *p* < 0.001). Colitis of grade 2 or higher was reported in 98% of patients with CRICAP compared to 78% of patients with ARICAP (*p* = 0.002). Loperamide and diphenoxylate/atropine were used more commonly in patients with ARICAP than in those with CRICAP (*p* < 0.001), and endoscopic APC was less frequently used for patients with ARICAP (16%) than for those with CRICAP (43%; *p* = 0.002), with 20% of delayed endoscopic evaluations requiring four sessions of APC. The median time from APC administration to recurrence among patients with recurrent symptoms was 4.8 months in the ARICAP group and 5.8 months in the CRICAP group.

### 3.3. Endoscopic Features

We stratified patients into early and delayed endoscopy groups based on the timing of the intervention (≤60 vs. >60 days after RICAP onset) (Table 3). In both groups, bleeding was the most common indication for endoscopy (71% and 52%, respectively), and neovascularization was the most common presentation, followed by bleeding. Lesions identified during the endoscopy were most commonly located in the rectum in both groups. Overall, 30% of patients who underwent endoscopy were treated with APC, and the number of repeated treatments was comparable in both groups.

The endoscopic findings for patients with RICAP consisted of edema, erythema, neovascularization, bleeding, ulceration, and stricture (Figure 1). The pathological characteristics of RICAP consisted of crypt architectural distortion (gland dropout), acute cryptitis, and abscess (Appendix A).

### 3.4. Univariate and Multivariate Analyses of Endoscopically Apparent Bleeding

Upon univariate analyses (Appendix A), the female sex (odds ratio (OR), 0.38; *p* = 0.015) and the use of GI-toxic chemotherapy (OR, 0.18; *p* < 0.001) were associated with a lower likelihood of having endoscopically apparent bleeding, whereas higher total RT doses (OR, 1.04; *p* = 0.013), longer RT durations (OR, 1.04; *p* = 0.032), and having CRICAP (OR, 3.41; *p* = 0.002) were associated with a higher likelihood of endoscopically apparent bleeding. In the multivariate analyses (Table 4), only the use of GI-toxic chemotherapy remained significantly associated with a lower risk of endoscopically apparent bleeding (OR, 0.26; *p* = 0.003).

### 3.5. Univariate and Multivariate Analyses of the Need for Medical Treatment

In the univariate analyses (Appendix A), older age (OR, 0.97; *p* = 0.060), higher total RT doses (OR, 0.94; *p* < 0.001) and numbers of RT sessions (OR, 0.88; *p* = 0.001), and having CRICAP (OR, 0.15; *p* < 0.001) were associated with a lower need for medical treatment. In contrast, the female sex (OR, 3.16; *p* = 0.006), NSAID use (OR, 4.62; *p* = 0.019), and the use of GI-toxic chemotherapy (OR, 3.18; *p* = 0.006) were associated with a higher need for medical treatment. In the multivariate analyses (Table 4), only the total RT dose (OR, 0.95; *p* = 0.007) and having CRICAP (OR, 0.23; *p* = 0.005) remained significantly associated with a lower need for medical treatment, whereas NSAID use continued to be significantly associated with a higher need for medical treatment (OR, 6.22; *p* = 0.014).

### 3.6. Univariate and Multivariate Analyses of the Need for Endoscopic APC and of the Recurrence of RICAP Symptoms

In the univariate analyses (Appendix A), the female sex (OR, 0.11; *p* < 0.001), the use of GI-toxic chemotherapy (OR, 0.29; *p* = 0.005), and the symptoms of abdominal pain (OR, 0.24; *p* = 0.014) and diarrhea (OR, 0.35; *p* = 0.024) were associated with a lower likelihood of undergoing endoscopic APC. In contrast, older age (OR, 1.10; *p* < 0.001), longer RT duration (OR, 1.08; *p* = 0.002), higher total RT dose (OR, 1.08; *p* < 0.001) and number of RT sessions (OR, 1.12; *p* = 0.002), bleeding symptoms (OR, 2.65; *p* = 0.036), and having CRICAP (OR, 3.99; *p* = 0.003) were associated with a higher need for endoscopic APC. In the multivariate analyses (Table 4), sex was the only factor that was associated with APC use (OR, 0.23; *p* = 0.039).

### 3.7. Subgroup Analysis by ARICAP and CRICAP Endoscopic Findings

Upon endoscopic evaluation, 8 out of the 51 patients with ARICAP had luminal ulcers (Appendix A). Endoscopy was usually performed earlier in those with ulcers than in those without (144 vs. 567 days; *p* = 0.005). RICAP recurrence was also more common in patients with ulcers than in those without (50% vs. 16%; *p* = 0.055). We also found that 45 patients with ARICAP had proctitis. Grade 2–3 colitis was more commonly seen in these patients than in those without proctitis (84% vs. 50%, *p* = 0.086), and those with proctitis received a longer duration of medical treatment than those without proctitis (18 vs. 7 days; *p* = 0.032).

Upon endoscopic evaluation, 12 out of the 61 patients with CRICAP had luminal ulcers (Appendix A). RICAP recurrence was more frequent in patients without ulcers than in those with ulceration (61.2% vs. 25%, *p* = 0.049). Fifty-three of the CRICAP patients had proctitis. Patients with proctitis, in comparison to those without proctitis, had a longer median RT duration (50 vs. 33 days; *p* = 0.002) and received a higher median RT dose (70 vs. 45 Gy; *p* = 0.001) over a higher median number of sessions (35 vs. 25; *p* = 0.034).

### 3.8. Clinical Outcomes

Overall, 34 patients in our cohort underwent APC (Appendix A). Patients who underwent APC had earlier endoscopy than those who did not undergo APC (4 vs. 9 months; *p* = 0.028), mainly for the indication of bleeding (91% vs. 42%; *p* < 0.001). Neovascularization was the most common finding during the endoscopy and was equally common in both groups. Patients who underwent APC had a higher rate of bleeding at the time of endoscopy (41% vs. 19%; *p* = 0.020), and lower rates of treatment with loperamide (18% vs. 49%, *p* = 0.003) and diphenoxylate/atropine (12% vs. 41%, *p* = 0.002). The rate of RICAP recurrence was not significantly decreased with APC treatment (50% vs. 35%; *p* = 0.144). Among the patients who underwent endoscopy due to clinically apparent bleeding, APC intervention did not decrease the recurrent bleeding rate either (*p* = 1.000). 

## 4. Discussion

The American Society of Colon and Rectal Surgeons’ guidelines for RICAP management focus on CRICAP rather ARICAP [9], as does the recent guideline for the use of APC issued by the American Society of Gastrointestinal Endoscopy (ASGE) [12]. Our study adds to the limited data on RICAP by analyzing the clinical features and outcomes of both subtypes. 

In our cohort, we noted several differences between ARICAP and CRICAP. First, the median time from RT to the development of ARICAP and CRICAP was 14 days and 13 months, respectively. The latter is slightly longer than previously reported (~8–12 months) [13]. Second, patients with CRICAP primarily presented with bloody diarrhea and colitis (grade ≥ 2), while most of those with ARICAP had non-bloody diarrhea. Bloody diarrhea was also the most common indication for endoscopy in both groups. Third, the proportion of patients who underwent medical treatment for their diarrhea was significantly lower in the CRICAP group compared to the ARICAP group. We hypothesize that this is because physicians are generally reluctant to treat patients with bloody diarrhea with antidiarrheal agents for fear of worsening colitis, which might then lead to more serious complications. Instead, endoscopic treatment was more frequently performed in patients with CRICAP.

In the endoscopy, most lesions were found in the rectum, as many cancer patients in our cohort were treated with pelvic RT and brachytherapy. We found that longer durations of radiotherapy with more sessions and higher doses of radiation were more closely associated with chronic inflammation more so than acute. The median RT dose administered to our cohort was 55 Gy (IQR, 50–70); such RT doses are known to cause more adverse effects than doses of less than 45 Gy, although they are associated with fewer longstanding injuries than RT doses greater than 70 Gy [14].

The proportion of patients with luminal ulceration found during the endoscopy was relatively small and comparable in the ARICAP and CRICAP groups. In patients who develop colitis after treatment with an immune-checkpoint inhibitor, luminal ulceration is highly suggestive of a more severe disease course, necessitating aggressive management and corticosteroid therapy [15]. However, luminal ulceration was not associated with a higher likelihood of RICAP recurrence in our study. The majority of patients had bleeding as the predominant symptom upon recurrence, with a big portion presenting with diarrhea. Ulceration is not part of the pathophysiology for either symptom, and these patients could very well not have ulceration, which would explain the observed result. Nonetheless, this may also be due to an underpowered sample size and warrants further study.

APC use was significantly greater in patients with CRICAP than in those with ARICAP. Delayed RT injuries are characterized by neovascularization and the emergence of leaky and fragile vessels [16]; this explains the predominantly bloody diarrhea observed in patients with CRICAP and, consequently, the need for APC. Neovascularization, however—previously deemed a characteristic feature of CRICAP—was also the most common reason for endoscopic APC in patients with ARICAP, even in the absence of significant clinical bleeding. Whether neovascularization in patients with ARICAP hints at the evolution of such lesions into a chronic process should be investigated further. This information might be useful in the early identification of lesions at higher risk of chronicity and worse outcomes and assist in screening and management strategies to prevent long-term RICAP sequelae. In addition, employing novel endoscopic technologies to classify neovascular RICAP lesions may help identify neovascular foci amenable to APC therapy.

The ASGE’s recently published recommendation on endoscopic management in patients with CRICAP is low-quality evidence and a conditional recommendation. In our study, we found that APC treatments performed in patients with bleeding did not reduce RICAP recurrences. Additionally, our finding that 20% of delayed endoscopic evaluations required four sessions of APC prompts questions about the optimal timing of and approach to endoscopic treatment of CRICAP. The body of evidence for endoscopic treatment of ARICAP is even weaker. As such, endoscopic evaluations and therapies should be used judiciously, ideally in patients with optimally dosed and timed trials of medical therapy, including suppositories/enemas. It should be noted that no APC-associated adverse events were reported for our cohort, highlighting the safety of this treatment modality. Further studies that will measure the efficacy of existing endoscopic treatments and explore new modalities for treating RICAP are imperative. Alternatively, topical formalin as a treatment of hemorrhagic RICAP has been reported as a safe strategy with comparable efficacy to APC [17]. However, the use of topical formalin remains limited in the clinical setting, and randomized trials comparing this therapy to APC are lacking. 

Most patients treated with antidiarrheal agents did not undergo APC. We speculate that the presence of hematochezia prompts endoscopy and may deprive patients of adequate antidiarrheal therapy, as is also apparent by the shorter median time from symptom onset to APC in these patients. Antidiarrheals may especially benefit patients who received high doses or numbers of sessions of RT. Previous studies have assessed short-chain fatty acid, sucralfate, and prednisone enemas for RICAP management [18]. A randomized, controlled trial showed that sucralfate enema therapy had a better clinical response compared to an oral sulfasalazine regimen in conjunction with prednisone enemas [19]. Additionally, sustained responses after the discontinuation of sucralfate enemas have been reported [20]. The role of systemic and topical immunosuppression or immunomodulation in RICAP remains controversial and requires further study.

The impact of chemotherapeutics with a notable GI toxicity profile on RT-associated RICAP has not been well studied. We found that while chemotherapy with GI toxicity did not contribute to either subtype of RICAP, it was associated with an increased need for medical therapy, less bleeding during endoscopy, and a lower rate of APC treatment. One possible explanation is that misattributing RICAP symptoms to chemotherapy ends with more medical management, which decreases the severity of symptoms and therefore bleeding and the need for APC. Our findings do not show synergism between chemotherapy and RT-associated adverse outcomes.

Being female was associated with more frequent use of medical therapy, less endoscopically obvious bleeding, and less use of APC. These findings further support the hypothesis that medical management may curb the progression of hematochezia and the need for endoscopic evaluation with APC therapy. Additionally, the more common use of medical therapy in females may reflect the typically broader multidisciplinary care with gynecological malignancies that receive care from a gynecologic oncology team. This warrants further study of a multidisciplinary approach to RICAP management.

Recent evidence suggests that the gut microbiome plays a critical role in gut inflammation and that intestinal dysbiosis has a detrimental effect on gut health. Fecal microbiota transplantation (FMT) could modulate the microbiome favorably and improve GI symptoms [21], as has been reported for patients with immunotherapy-related colitis and CRICAP in small case series [22]. Large-scale studies are needed to further explore the role of the gut microbiome and the value of FMT in RICAP.

Our study has limitations. First, this retrospective study did not account for RT fractionation schedules, which may have affected disease courses. Second, limiting the inclusion criteria to those who had an endoscopy may have led to selection bias with patients who have more severe disease. Third, the lack of pre-therapy endoscopy in most patients suggests that the features identified during endoscopy after RT may have been present prior to RT. Finally, since most patients had proctopathy, the sample size of patients with non-rectal injuries may have been underpowered.

## 5. Conclusions

RICAP is a well-recognized adverse complication of RT, wherein diarrhea and hematochezia are common clinical presentations of ARICAP and CRICAP, respectively. Medical management of ARICAP and CRICAP should be implemented before endoscopic therapies are considered. APC may be useful for RICAP patients with bleeding, but further research into its impact on recurrent bleeding is needed. Future, larger, well-designed studies are needed to validate our findings and hypotheses.

## Figures and Tables

**Figure 1 cancers-15-03160-f001:**
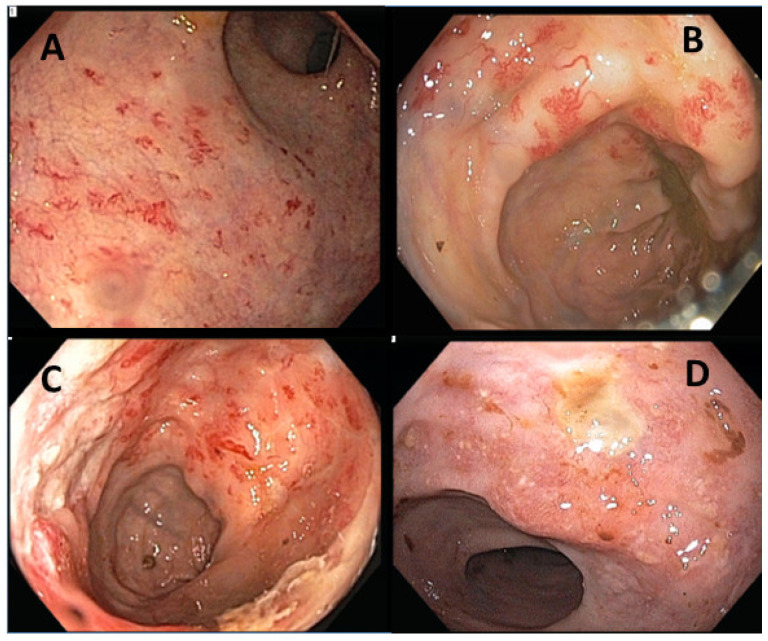
Endoscopic presentations of RICAP. (**A**–**C**) show neovascularization of the colonic mucosa with spontaneous bleeding. (**D**) shows mucosal ulceration with edema and erythema.

**Table 1 cancers-15-03160-t001:** Demographic information of patients with RICAP, *n* = 112.

Characteristics	No. of Patients
Median age, years (IQR)	59 (50–68)
Male sex (%)	50 (45)
Race/ethnicity (%)	
White	94 (84)
Black	9 (8)
Hispanic	5 (4)
Asian	4 (4)
Comorbidities (%)	
Hypertension	43 (38)
Diabetes	16 (14)
Heart disease	13 (12)
Other	11 (10)
Smoking (%)	61 (55)
NSAID use (%)	24 (21)
Cancer type (%)	
Prostate	37 (33)
Recto-sigmoid	23 (21)
Anal	24 (21)
Cervical	17 (15)
Miscellaneous	11 (10)
Cancer stage (*n* = 98) (%)	
I	37 (33)
II	34 (30)
III	21 (19)
IV	6 (5)
GI system-toxic chemotherapy within 6 months of radiotherapy (%) ^a^	56 (50)
Radiotherapy type (%)	
External beam ^b^	92 (82)
Brachytherapy ^c^	20 (18)
Median number of radiotherapy sessions (IQR)	30 (26–35)
Median length of radiotherapy, days (IQR)	43 (37–51)
Median dose of radiotherapy, Gy (IQR)	55 (50–70)
Non-GI adverse events (%)	
Skin	30 (27)
Urinary system	6 (5)
Other	7 (6)
None	69 (62)
Median follow-up duration, years (IQR) ^d^	10 (5–13)

^a^ GI-toxic chemotherapy includes drugs from the following classes that have been reported in the literature: tyrosine kinase inhibitors, immune checkpoint inhibitors, taxanes, platinum-based regimens, nitrosoureas, alkylating agents/anti-metabolites, and anthracyclines. ^b^ The external beam radiation group included patients who received intensity-modulated radiation therapy (45), 2- or 3-dimensional conformational radiation therapy (8), proton therapy (6), photon therapy (6), or electron therapy (3). Detailed information was missing for 24 patients. ^c^ The brachytherapy group included patients who received monotherapy (3) and combination therapy (17). ^d^ The follow-up duration was the time from colitis onset to death or the date of the last encounter. Abbreviations: GI, gastrointestinal; IQR, interquartile range; NSAID, nonsteroidal anti-inflammatory drug; RICAP, radiation-induced colitis and proctopathy.

**Table 2 cancers-15-03160-t002:** Comparison of the clinical characteristics of patients with acute vs. chronic RICAP.

Characteristics	ARICAP *n* = 51	CRICAP *n* = 61	* p * Value
Duration of radiotherapy, days, median (IQR)	41 (37–43)	49 (36.5–55.0)	0.009
Number of radiotherapy sessions, median (IQR)	28 (25–30)	35 (28–38)	<0.001
Dosage of ionizing radiation, Grey, median (IQR)	54 (45–58)	66 (50.4–75.6)	<0.001
Median duration from first radiotherapy dose to RICAP, days (IQR) ^a^ *n* = 112	14 (7–21)	404 (183–689)	<0.001
Clinical symptoms (%)			
Diarrhea	47 (92)	38 (62)	<0.001
Blood in stool	12 (24)	56 (92)	<0.001
Nausea and vomiting	23 (45)	1 (2)	<0.001
Abdominal/rectal pain	20 (39)	12 (20)	0.035
Median duration of symptoms, days (IQR) (*n* = 94)	16 (8–23)	29 (16–74)	<0.001
Hospitalization (%)	9 (18)	7 (12)	0.421
ICU treatment (%)	1 (2)	1 (2)	1.000
Colitis, grade 2 or higher (%)	40 (78)	60 (98)	0.002
Diarrhea grade (*n* = 64) (%)			0.831
1	12 (28)	7 (33)	
2	22 (51)	9 (43)	
3	9 (21)	5 (24)	
RICAP medical treatment (%)			
Loperamide	36 (71)	8 (13)	<0.001
Diphenoxylate/atropine	26 (51)	10 (16)	<0.001
Suppository/enema ^b^	11 (28)	13 (21)	1.000
Topical corticosteroid	4 (8)	4 (7)	1.000
Intravenous fluid	6 (1)	7 (12)	1.000
Median duration of medical treatment, days (IQR) (*n* = 63)	16 (8–26)	23 (10–36)	0.255
Endoscopic APC treatment (%)	8 (16)	26 (43)	0.002
Overall recurrence of clinical symptoms (%)	30 (59)	43 (71)	0.234
Median time from symptom onset to first recurrence, months (IQR) (*n* = 73)	12 (8–25)	15 (4–75)	0.783
Median time from APC treatment to recurrence, months, median (IQR) ^c^ *n*_1_ = 2; *n*_2_ = 13	4.8 (2.9–6.7)	5.8 (2.5–28.6)	-

^a^ The median duration from the first dose of radiotherapy to colitis onset was used for comparison. The patients with ARICAP had a median duration comparable to that of the overall radiotherapy course because most patients (49) experienced colitis onset between the first and last doses of radiotherapy. ^b^ Non-steroidal enemas/suppository types included mesalamine, short-chain fatty acid, sucralfate, and combined regimen. ^c^ A total of 25 patients receiving APC had symptom recurrence (colitis, with or without re-bleeding), 7 of which were from the ARICAP group, and 18 were from the CRICAP group. Of these, 15 had symptom recurrence after APC (the rest received APC because of recurrent symptoms), 2 from the ARICAP group, and 13 from the CRICAP group. Statistical analysis was not performed given this small sample size. Abbreviations: APC, argon plasma coagulation; ARICAP, acute radiation-induced colitis and proctopathy; CRICAP, chronic radiation-induced colitis and proctopathy; ICU, intensive care unit; IQR, interquartile range; RICAP, radiation-induced colitis and proctopathy.

**Table 3 cancers-15-03160-t003:** Comparison of the clinical characteristics of patients who received early vs. delayed endoscopies ^a^.

Characteristics	Early Endoscopy * n * = 31	Delayed Endoscopy * n * = 81	* p * Value
Median days between initial symptom to endoscopy (IQR) (*n* = 112)	13 (1–30)	331 (151–811)	<0.001
Symptoms leading to endoscopy (%)			
Bleeding	22 (71)	42 (52)	0.088
Diarrhea	4 (13)	10 (12)	1.000
Abdominal pain	4 (13)	9 (11)	0.751
Others ^b^	4 (13)	26 (32)	0.055
Endoscopic presentation (%)			
Neovascularization	23 (74)	60 (74)	0.306
Bleeding	8 (26)	21 (26)	1.000
Ulceration	5 (16)	15 (19)	1.000
Stricture	2 (7)	2 (3)	1.000
Location (%)			1.000
Rectum	27 (87)	71 (88)	
Rest of the colon	4 (13)	10 (12)	
Endoscopic APC treatment ^c^ (%)	11 (36)	23 (28)	0.496
Repeat treatment	5 (16)	10 (12)	0.757
Number of treatments			0.776
2	3 (60)	6 (60)	
3	2 (40)	2 (20)	
4	0 (0)	2 (20)	

^a^ The indications for endoscopy in these patients included routine screening, follow-up after polyp or cancer, procedural intervention for other concerns, and investigation of presenting symptoms of diarrhea, bleeding, and abdominal or rectal pain. ^b^ Others included nausea, vomiting, obstruction, and surveillance. ^c^ APC was performed only for vascular lesions. Abbreviations: APC, argon plasma coagulation; IQR, interquartile range.

**Table 4 cancers-15-03160-t004:** Multivariate analyses.

Covariate	Odds Ratio (95% Confidence Interval)	*p* Value
Endoscopically apparent bleeding		
GI-toxic chemotherapy	0.26 (0.11–0.63)	0.003
Length of radiotherapy	1.02 (0.97–1.06)	0.439
CRICAP vs. ARICAP onset	1.99 (0.83–4.77)	0.121
Need for medical treatment		
NSAID	6.22 (1.45–26.65)	0.014
Dose of radiotherapy	0.95 (0.91–0.99)	0.007
CRICAP vs. ARICAP onset	0.23 (0.08–0.64)	0.005
Need for endoscopic treatment		
Female vs. male	0.23 (0.06–0.93)	0.039
Length of radiotherapy	1.03 (0.96–1.11)	0.445
Dose of radiotherapy	1.09 (0.97–1.23)	0.148
Number of radiotherapy sessions	0.80 (0.63–1.03)	0.081
CRICAP vs. ARICAP onset	1.68 (0.54–5.20)	0.371

Abbreviations: ARICAP, acute radiation-induced colitis and proctopathy; CRICAP, chronic radiation-induced colitis and proctopathy; GI, gastrointestinal; NSAID, nonsteroidal anti-inflammatory drug.

## Data Availability

Data can be made available upon request by contacting the corresponding author.

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
