# Peer review of "Clinical Features and Management of Acute and Chronic Radiation-Induced Colitis and Proctopathy"

_cancers, 2023, doi:10.3390/cancers15123160_

Round 1
Reviewer 1 Report
The authors investigated radio-induced colitis and proctopathy (RICAP) in this retrospective study. The topic is of particular importance because these radio-induced side effects can have a more or less serious impact on the quality of life of the patients. Doctors need to be able to propose the most appropriate treatments to help them. This requires identifying clinical characteristics and outcomes of RICAP. This study should provide some data on this objective.
The study is well structured and described. A total of 112 patients with confirmed RICAP were enrolled in the study. The data (Tables and figure) and the statistical analysis are very detailed and allow the reader to make sense of the results.
Author Response
Thank you for your positive review. We agree that there is an important need to explore the nuances of RICAP presentation, treatment, and outcomes.
Reviewer 2 Report
This work is dealing with RICAP general but most important information about the dose to rectum and proctus is lacking. If they do focus on the treatment of RICAP not referring to radiation details can be permissible, but they are talking about incidence of RICAP. We have already many amount of RICAP after pelvic irradiation mainly in radiation therapy related journals, but they were not referred at all. Also the indication of endoscopic exam is not clear. We do not know what drug is bowel toxic, please define it. There are patients who have acute as well as chronic RICAP both. How did you treat those patients? There are so many things that must be improved.
If you concentrate your work only to the endoscopic treat,ent and its efficacy the work could be readable to us.
Author Response
Response: Thank you for your detailed feedback and advice on our manuscript. To make it easier, we will respond in a point by point fashion:
- Important information about the dose to rectum and proctus is lacking. If they do focus on the treatment of RICAP not referring to radiation details can be permissible, but they are talking about incidence of RICAP. We have already many amount of RICAP after pelvic irradiation mainly in radiation therapy related journals, but they were not referred at all.
Response: Our study aims to mainly focus on the clinical aspect of RICAP from a GI standpoint (presentation, treatment, outcomes), we chose not to include detailed information regarding radiotherapy parameters. We agree that this is important information, but it is not the major interest of our paper, nor are we trying to suggest an incidence of RICAP. In the results, when we mention that 112 of the initial 61,399 patients developed RICAP, we are only describing our patient selection process, not incidence. However, as the reviewer suggested, we added a brief section in the introduction to describe incidence and radiotherapy-related risk factors for RICAP, a sentence in the methods to explain that our focus is purely on the clinical aspect of RICAP, and additional analysis to show the difference in radiotherapy duration and dosage between ARICAP and CRICAP groups.
- Also the indication of endoscopic exam is not clear. We do not know what drug is bowel toxic, please define it.
Response: Added footnotes to tables 3 and 1 to address these concerns, respectively (table 3 – indications for endoscopy; table 1 – description of GI-toxic chemotherapy).
- There are patients who have acute as well as chronic RICAP both. How did you treat those patients?
Response: In our study, we classified patients as either acute RICAP or chronic RICAP depending on the time from last radiotherapy session to symptom onset, meaning no patients were considered as having both. In real-world practice, some patients may certainly present with acute on chronic RICAP, and in these cases we would treat the predominant symptoms that are troubling the patient.
We have made many edits to the manuscript and tables, and we hope that with these improvements you will find our work more satisfactory.
Reviewer 3 Report
This manuscript is an original article that retrospectively analyzed the clinical and endoscopic characteristics of acute or chronic radiation-induced colitis and proctopathy. The authors showed that neovascularization and bleeding was the most frequent finding on endoscopy. Furthermore, the authors found that APC treatment did not significantly reduce bleeding recurrence or RICAP symptoms although CRICAP patients with bleeding more often received argon plasma coagulation.
This study was conducted well, the methods are appropriate, and the data are presented clearly. I think this is a well-written paper with interesting data.
The results will be of interest to clinicians and researchers in the field.
The following a minor issue require clarification:
Minor
1. (P2L75) “Gastrointestinal” should be abbreviated to “GI”.
2. Please provide detailed information regarding GI system-toxic chemotherapy.
3. (P11L226) The authors found that RICAP recurrence was more frequent in patients without ulcers than in those with ulceration. Please discuss this contradict result in the Discussion section.
4. I recommend that the authors provide the data of the duration from APC treatment to recurrence in the patients with bleeding.
Author Response
Response: Thank you very much for your kind feedback and valuable input. We have incorporated your suggestions as below:
The following a minor issue require clarification:
Minor
- (P2L75) “Gastrointestinal” should be abbreviated to “GI”.
Fixed.
- Please provide detailed information regarding GI system-toxic chemotherapy.
Added a footnote to table 1 detailing the different classes of chemotherapy agents that are considered GI-toxic.
- (P11L226) The authors found that RICAP recurrence was more frequent in patients without ulcers than in those with ulceration. Please discuss this contradict result in the Discussion section.
This was an interesting finding and we suppose that the reason for recurrence depends on the presenting symptom upon recurrence. For instance, the underlying pathology in patients whose recurrence presents as bleeding is likely neovascularization. The pathology in patients presenting with diarrhea likely have underlying proctitis with or without ulcers. As such, ulcers may not be a key part of the pathophysiology for recurrence, especially in our sample where most recurrent cases presented with bleeding (~50/72 cases) versus 26/72 with diarrhea and only 15/72 with abdominal or rectal pain. We included a brief section in the discussion to explain this.
- I recommend that the authors provide the data of the duration from APC treatment to recurrence in the patients with bleeding.
Added this information to table 2 as well as the results.
Round 2
Reviewer 2 Report
Surely the authors concentrated only onto the symptoms and symtom management of radiation injury to rectum and anus. For the contribution factor of the injury, intestinal toxic agents were referred but the results would be changed if the authors also include another radiation related factors in multivariate analysis.